# Joint Risk Analysis of Extreme Rainfall and High Tide Level Based on Extreme Value Theory in Coastal Area

**DOI:** 10.3390/ijerph20043605

**Published:** 2023-02-17

**Authors:** Hao Chen, Zongxue Xu, Ji Chen, Yang Liu, Peng Li

**Affiliations:** 1College of Water Sciences, Beijing Normal University, Beijing 100875, China; 2Beijing Key Laboratory of Urban Hydrological Cycle and Sponge City Technology, Beijing 100875, China; 3Department of Civil Engineering, The University of Hong Kong, Hong Kong 999077, China; 4College of Geoscience and Surveying Engineering, China University of Mining and Technology, Beijing 100875, China

**Keywords:** joint risk, urban flood disaster, extreme rainfall, high tide level, bivariate copula

## Abstract

Extreme rainfall and high tide levels are the main causal factors of urban flood disasters in coastal areas. As complex interactions between these factors can exacerbate the impact of urban flood disasters in coastal areas, an associated flood risk assessment involves not only the estimation of the extreme values of each variable but also their probability of occurring simultaneously. With a consideration of the Shenzhen River Basin (China), this study used bivariate copula functions to quantitatively evaluate the joint risk of extreme rainfall and a high tide level. The results showed that a significant positive correlation exists between extreme rainfall and the corresponding high tide level, and that if the positive dependency was ignored, the probability of simultaneous extreme events would be underestimated. If a dangerous event is defined as one in which heavy rainfall and high tide level events occur concurrently, the “AND” joint return period based on the annual maxima method should be adopted. If a dangerous event is defined as one in which either only a heavy rainfall or a high tide level event occurs, the “OR” joint return period should be adopted. The results represent a theoretical basis and decision-making support for flood risk management and flood prevention/reduction in coastal areas.

## 1. Introduction

Flooding, which is one of the most frequent types of natural disaster globally, can have serious socioeconomic implications. With recent global climate changes and the acceleration of urbanization processes, urban flood disasters are becoming increasingly serious, and represent a prominent problem for urban community security and socioeconomic development [1,2]. The causes of urban flood disasters are complex. In coastal areas, urban flood disaster events are not only closely related to precipitation, topography, urban flood protection and waterlogging drainage engineering, urban management and other factors, but are also strongly affected by tide level. Moreover, the combination of extreme precipitation and a high tide level occurring simultaneously or in close succession will aggravate the severity of urban flood disasters in coastal areas, potentially resulting in even greater economic losses and casualties [3,4,5]. Therefore, a quantitative assessment of the dependence and probability of encountering extreme precipitation and a high tide level could provide important decision-making support for flood protection as well as waterlogging drainage planning and design in coastal cities.

Owing to the complex interactions between precipitation and tides in coastal areas, the traditional single-variable hydrological frequency method, which selects only one characteristic for analysis, cannot satisfy the requirements of current flood risk assessments. Therefore, multivariate joint analysis methods that include two or more hydrological variables have been recently incorporated into the analysis and calculation of such flood risks. Bivariate probability distribution models mainly include the bivariate normal distribution model, bivariate gamma model, meta-Gaussian distribution model, and Farlie–Gumbel–Morgenstern (FGM) model [6,7,8,9]. However, such models are based on the assumption of linear dependence between variables, which means they have difficulty describing nonlinear and asymmetric random variables. Moreover, some models assume that the variables obey the same marginal distribution, whereas the variables do not obey the same probability distribution in most cases. The copula functions developed by Sklar (1959) overcome the shortcomings of the above models [10]. They have an arbitrary marginal distribution and can describe the nonlinear and asymmetric dependence between variables. Owing to their flexibility and efficiency, copula functions have received considerable attention and have been applied widely in the field of hydrology.

The existence of statistical dependence between extreme hydrological events has been recognized for a long time, and a lot of studies have been conducted to attempt to quantify its strength. For example, following a statistical analysis of 628 pairs of tide–streamflow data from the Rappahannock River in Chesapeake Bay (USA), Loganathan et al. (1987) reported the existence of such a dependence between high flows and high tides [11]. An independent estimation of the joint probability of the simultaneous occurrence of flows and tides led to their underestimation. Coles et al. (1999) quantified the statistical dependence between a rainfall series and wave–surge levels in southwest England [12]. Svensson and Jones (2002) investigated the statistical dependence between high sea surges, river flows and precipitation in eastern Britain [13]. This study found that the strongest flow–surge dependence occurred between the river flow on the north shore of the Firth of Forth and a sea surge at Aberdeen, Wick, and Lerwick, and that if one variable exceeded a certain return period, then the risk of the other variable exceeding the same return period was 20%. The dependence between river flow and surge was found to be stronger during winter than summer, and a lagged analysis revealed that the dependence was strongest when flow and surge occurred on the same day. Svensson and Jones (2004) performed a similar study with daily resolution in southern and western Britain. They reported that a statistically significant dependence existed between river flow and daily maximum sea surge for catchments spread along most of this coastline, and that the strength of the dependence was governed by a range of factors that included meteorological conditions, the orographic properties of the catchment, soil moisture, cyclones, and lag time [14].

The application of copula functions for bivariate frequency analysis in hydrology was first proposed by Favre et al. (2004), who quantified the dependence between annual peak flows and volumes in the Rimouski River (Canada) using the Frank and Clayton copula [15]. Grimaldi and Serinaldi (2006) built a trivariate joint distribution of flood event variables using the asymmetric Archimedean copula functions, and reported that the three variables (peak, volume, and duration) were correlated in several manners that were dependent on the threshold used to identify the flood event [16]. Recently, many studies have used the copula model to quantify the combined effect of rainfall and tide processes in coastal areas. For example, both Wahl et al. (2015) and Zellou and Rahali (2019) used a copula-based model to explore the joint probability and effect of extreme rainfall and storm surge in the United States and Morocco, respectively [17,18]. They confirmed that a significant dependence existed between these two hydrological variables. Moreover, they reported that the number of compound events has increased considerably over the past century, and suggested that this trend might continue under the scenario of a changing environment. In China, several similar studies have also been conducted [19,20,21,22]. Both Lian et al. (2013) and Xu et al. (2014) investigated the joint probability of extreme precipitation and storm tides in the city of Fuzhou (Fujian Province, China), and evaluated its change using copula-based models [19,20]. They reported the existence of a certain positive dependence between rainfall and tidal level, and indicated that the joint probability had increased by more than 300% on average since 1984. Tu et al. (2018) studied the dependence of heavy precipitation and high tidal level in Shenzhen (Guangdong Province, China), and confirmed the effectiveness of the Archimedean copula in modeling the joint distribution of precipitation and tides [21]. Xu et al. (2019) used copula functions to investigate the bivariate return period of compounding rainfall and storm tide events in Haikou (Hainan Province, China) [22]. They reported a significant correlation between rainfall and storm tides, and found that the analysis of the bivariate return period provided information about risk that was more adequate and comprehensive than that provided by an analysis of the univariate return period.

However, for different coastal regions, the copula functions and marginal distribution functions of rainfall and tide level data samples obtained from different sampling methods are different. How to select the best-fit marginal distribution functions and copula functions is important for the assessment of the joint risk of rainfall and tide in coastal areas. In addition, flood and tide prevention facilities in coastal cities are usually designed to resist disasters with a specific return period (e.g., 50 years or 100 years, etc.). The bivariate return period of rainfall and storm surge is usually described by “AND”, “OR”, and the Kendall return period. However, which one is the safer design standard for urban flood and tide prevention facilities? It is of great significance to solve these problems for flood and tide risk management and the design of flood and tide prevention facilities in coastal cities [23,24,25].

Therefore, this study used the annual maximum method and the peaks over threshold sampling method to obtain the samples of extreme rainfall and high tide level, selected the best-fit marginal distribution functions and copula functions through different test methods, quantitatively evaluated the joint risk of extreme rainfall and high tide level in the Shenzhen River Basin on this basis, and analyzed the applicability of different types of joint return period. The results provide a theoretical basis and decision-making support for flood/waterlogging risk management, but could also be used as reference for flood prevention/reduction measures, and other applications in relation to other coastal cities.

The remainder of this paper is organized as follows: The case study area, the Shenzhen River Basin (China), and the rainfall and tide data series are presented in Section 2. Section 3 describes the methods adopted in this study, including extreme value sampling methods, bivariate copula functions, and the different types of joint return period. The results of the joint risk of extreme rainfall and high tide level obtained from the two sampling methods are reported and discussed in Section 4. Finally, the derived conclusions are described in Section 5.

## 2. Study Area and Data Description

The Shenzhen River Basin is located in the east of the Pearl River Estuary (Southeast China), which is an important component of the Guangdong–Hong Kong–Macao Greater Bay Area. The catchment area is bordered by mountains to the north, east, and south and by Shenzhen Bay in the west. The northern part of the catchment area belongs to the city of Shenzhen and the southern part belongs to the Hong Kong Special Administrative Region, with a total area of approximately 312.5 km^2^ (Figure 1). The Shenzhen River originates near Niuweiling in the Wutong Mountains and flows southwestward for approximately 33.4 km into Shenzhen Bay. Its major tributaries include the Shawan, Buji, and Futian rivers on the Shenzhen side and the Pingyuan, Wutong, and Xintian rivers on the Hong Kong side. Given the maritime subtropical monsoon climate, the mean annual precipitation of the Shenzhen River Basin is approximately 1764.14 mm.

The socioeconomic development of the Shenzhen River Basin is vital for both Shenzhen and Hong Kong. However, population and wealth have gathered rapidly with the rapid development of urbanization in urban areas, which makes urban areas more sensitive and vulnerable to floods. Frequent flood disasters seriously hinder sustainable regional development. Despite an increase over recent decades in the number of flood controls and drainage projects in the Shenzhen River Basin, and the gradual improvement of emergency management measures, occurrences of urban flood disasters continue to increase. The principal root causes of this are as follows: (a) the intensity and frequency of extreme precipitation have increased under the influence of climate change, even though the average annual precipitation has changed little; (b) rapid urbanization has increased the impervious area of the catchment, meaning that the runoff-yield time has accelerated, runoff-yield flow has increased, and convergence time has shortened; and (c) the sea level rise due to global climate change means that the tide level in Shenzhen Bay has become elevated. An extreme high tide is able to directly cause a severe urban flood disaster. Additionally, during a period of extreme high tide, the flood of the Shenzhen River cannot be discharged, which causes the water level of the river to rise. In this circumstance, the drainage capacities of the urban pipe network are greatly reduced, which can lead to a worse situation of backward flow, increasing the risk of an urban flood disaster. Especially when extreme precipitation coincides with high tide level, the severity of an urban flood disaster in the Shenzhen River Basin can be greatly exacerbated. Therefore, instead of considering only heavy precipitation, the joint distribution of precipitation and tide should be considered in relation to urban flood control and drainage planning/design.

This study used daily precipitation data from the Shenzhen Reservoir rainfall station and tide data from the Chiwan tide station (data source: Shenzhen Hydrological Data Almanacs). On the basis of the length of the observational data, the time series selected covered the period 1965–2017. Owing to the different datums adopted in the process of compiling the almanacs for the selected tide data, the value of 2.463 m below the Pearl River datum was selected in this study as the unified datum for processing.

## 3. Method

### 3.1. Methods for Sampling Extreme Values

The annual maxima sampling method and peaks over threshold sampling method are usually used to select extreme values of precipitation and tide series data [26,27,28]. The annual maxima method selects only the annual maximum daily rainfall or high tide level as samples, which is considered better and safer for engineering design. However, in data-sparse areas, only a few samples could be selected using the annual maxima method, which would be unsuitable for statistical analysis. Moreover, in coastal areas, several extreme rainfall events might occur annually. For the encounter of extreme rainfall and high tide level, the annual maxima method might omit other extreme rainfall events that are lower than the annual maximum but that might encounter a higher tide level. The peaks over threshold sampling method could obtain a greater number of samples of extreme rainfall and high tide level by taking the daily rainfall or tide level of a certain percentile as the threshold. Taking the rainfall series as an example, the specific steps are as follows: rank the effective daily rainfall (>0.1 mm) of many years from small to large, take the daily rainfall of a specific percentile as the extreme rainfall threshold, and set it as the extreme rainfall day if the daily rainfall exceeds the threshold.

Two methods for selecting a sample for the encounter design of extreme rainfall and high tide level are available: priority of precipitation and priority of tide. The former process samples the daily rainfall series first and then selects the corresponding high tide level of the same day. The latter process samples the daily high tide level series first and then selects the corresponding daily precipitation of the same day. For sampling based on the priority of tide, it is generally observed that the daily high tidal levels selected by the annual maxima and peaks over threshold methods frequently meet no rainfall event. Therefore, in this study, the annual maxima method and peaks over threshold method (95% threshold, 64.5 mm) were used to sample the rainfall series first, and the high tide level was then selected based on the priority of rainfall.

### 3.2. Copula Functions and Marginal Distributions

Copulas are a type of distribution function that join multivariate distribution functions to their one-dimensional marginal distribution functions, and they have emerged as a powerful approach for the simplification of multivariate stochastic analysis [29]. According to the theorem of Sklar (1959), if *F* is an *n*-dimensional distribution function, with marginal distributions *F_1_*, *F_2_*, …, *F_n_*, then there exists an *n*-copula function *C* such that:(1)$$F(x_{1},x_{2}\ldots ,x_{n})=C(F_{1}(x_{1}),F_{1}(x_{2}),\ldots ,F_{n}(x_{n}))\;\;x_{1},x_{2},\ldots ,x_{n}\in R_{n}$$
where *C* is unique when *F_1_*, *F_2_*, …, *F_n_* are continuous marginal distributions.

Copula functions can be divided into three types: ellipsoid, Archimedean, and quadratic. Of these, only Archimedean copula functions contain one parameter associated with the Kendall rank correlation coefficient, and thus they are used widely to establish a joint distribution of hydrological variables. In this study, common bivariate Archimedean copula functions (i.e., the Gumbel Copula, Clayton Copula, and Frank Copula) were compared to select the best-fit copula function for the joint distribution of extreme rainfall and tides. The details of the three generators, cumulative distributions, relationships between parameter and Kendall rank correlation coefficient, and parameter ranges are listed in Table 1. The dependence of extreme rainfall and tides was also quantified using the Pearson correlation coefficient and Spearman rank correlation coefficient.

Moreover, five distributions commonly used in the hydrological field (i.e., the 3-parameter univariate generalized extreme value (GEV) distribution, and four two-parameter univariate distributions, i.e., the Weibull distribution, lognormal distribution, gamma distribution, and loglogistic distribution) were applied as the marginal distribution to fit the extreme rainfall and tides.

### 3.3. Goodness-of-Fit Test of Copula and Marginal Distribution Functions

The most important issues in application of copula functions to analyze the dependence of extreme rainfall and tides are the selection of the optimal bivariate copula and the univariate marginal distribution function. Therefore, it is necessary to test the fitting precision of different marginal distribution functions and copulas. In this study, the Kolmogorov–Smirnov (KS) test and Cramér–von Mises (CvM) test were used for the test of distance to avoid large deviations between the empirical distribution function and fitted distribution function [30,31]. Akaike information criteria (*AIC*) and Bayesian information criteria (*BIC*) were used for the overfitting test to avoid the false appearance of overfitting of the fitted distribution function [32,33,34]. The KS statistic *D* and CvM statistic $W_{n}^{2}$ can be computed as follows:(2)$$D=\sup\left|F_{n}(x)-F(x)\right|$$
(3)$$W_{n}^{2}=n\int_{-\infty }^{\infty }{\left(F_{n}(x)-F(x)\right)}^{2}dx$$
where $F_{n}(x)$ and F(x) are the empirical distribution function and fitted distribution function, respectively, and *n* is the number of observations.

The formulas for computing the *AIC* and *BIC* are as follows:(4)AIC=2k−2ln(L)
(5)BIC=kln(n)−2ln(L)
where *k* is the number of parameters in fitted distribution function, and *L* is the likelihood function.

### 3.4. Joint Return Period

The hydrological return period refers to the average interval of time between two consecutive occurrences of a hydrological variable that exceed or equal a certain threshold [35]. In the applications, especially in the fields of environmental sciences and hydraulic engineering, the return period of a certain hydrological variable is usually regarded as a standard of engineering design. In most situations, the analysis and application of a return period involve only a single hydrological variable. However, certain disaster events, such as flooding and waterlogging, might be caused by two or more nonindependent hydrological variables [36,37]. Therefore, it is necessary to conduct a bivariate or multivariate frequency analysis to avoid the underestimation of disaster risk attributable to univariate frequency analysis.

In coastal areas, an urban flood and waterlogging disaster might occur when both hydrological variables of rainfall R and tide T exceed certain respective threshold simultaneously, or when only one of the variables exceeds its threshold. The former joint probability distribution is defined as “AND”, while the latter joint probability distribution is defined as “OR”. The joint probability distributions corresponding to these two events are defined as follows:(6)$$E_{AND}=P((R>r)\cap (T>t))=1-F_{r}(r)-F_{t}(t)+F_{c}(r,t)$$
(7)$$E_{OR}=P((R>r)\cup (T>t))=1-F_{c}(r,t)$$

Therefore, the joint return periods are denoted as follows:(8)$$T_{AND}=\frac{1}{M\cdot E_{AND}}$$
(9)$$T_{OR}=\frac{1}{M\cdot E_{OR}}$$
where $F_{r}(r)$ and $F_{t}(t)$ are the marginal distributions for extreme rainfall and high tide level, and $F_{c}(r,t)$ is the joint distribution function. *M* is the annual mean number of extreme rainfall events. If *N* is the number of years of observational data and *n* is the number of extreme rainfall events, then *M* is denoted as follows:(10)$$M=\frac{n}{N}$$

In this study, for the annual maxima method, *M* = 1; for the peaks over threshold sampling method, *M* = 6.45.

The traditional “AND” and “OR” joint return periods cannot accurately identify dangerous events. In “AND” cases, a dangerous event whose joint probability is greater than a given joint probability level might be regarded as a safe event; in “OR” cases, a safe event whose joint probability is less than a given joint probability level might be regarded as a dangerous event [38]. To resolve the above problems, the Kendall’s measure function *Kc* was introduced to improve the calculation of the traditional joint return period [39], which expresses the probability that the joint distribution probability is less than a given probability level [40]. Thus, *Kc* can be defined as follows:(11)$$K_{c}(t)=P(C(\mu ,\nu )\leq t)$$
where *t*∈(0,1) is a critical probability level. Therefore, the Kendall’s return period, based on *Kc*, can be defined as follows:(12)$$T_{k}=\frac{1}{M(1-P(C(\mu ,\nu )\leq t))}=\frac{1}{M(1-K_{c}(t))}$$

For an Archimedean copula, *Kc(t)* can be calculated using the following equation:(13)$$K_{c}(t)=t-\frac{\varphi (t)}{\varphi^{\prime }(t^{+})}$$
where $\varphi \left(t\right)$ is the generating function of the Archimedean copula, and $\varphi^{\prime }\left(t^{+}\right)$ is the right derivative of $\varphi \left(t\right)$.

## 4. Results and Discussion

### 4.1. Statistical Analysis of Extreme Rainfall and Correspondent Tide Level

Using daily rainfall data from the Shenzhen Reservoir rainfall station and tide data from the Chiwan tide station acquired in the period 1965–2017, this study adopted the annual maxima and peaks over threshold sampling methods to sample the rainfall series, and the extreme precipitation and corresponding high tide series are shown in Figure 2. It can be seen that the number of samples obtained using the annual maxima sampling method is 53, the range of extreme rainfall is 89.0–338.5 mm, and the corresponding range of the high tide level is 3.33–4.08 m. For the peaks over threshold sampling method, the number of samples obtained is 342, the occurrence is 6.45 times that of the annual mean, the range of extreme precipitation is 64.5–338.5 mm, and the corresponding range of high tide level is 3.272–5.027 m.

The fundamental statistical characteristics of the extreme rainfall and corresponding high tide series obtained using the annual maxima and peaks over threshold methods are illustrated in Figure 3. For the rainfall series, the maximum values of the two sampling methods are consistent, whereas the median values, minimum values, and ranges of normal values have significant differences. The median and minimum values of the annual maxima method are 158.3 and 89 mm, respectively, whereas the median and minimum values of the peaks over threshold method are obviously lower, that is, 87 and 64.5 mm, respectively. The range of normal values of the annual maxima method is 89.0–308.5 mm, and only two samples fall outside the upper boundary of the normal values. The range of normal values of the peaks over threshold method is smaller, that is, 64.5–183.2 mm, and 23 samples fall outside the upper boundary of the normal values. It shows that for rainfall samples using peaks over threshold method, the magnitude and range of normal values significantly decreased and more higher values were recognized to be abnormal.

For the corresponding high tide level series, the differences between the median values, minimum values, and ranges of normal values of the two sampling methods are reasonably small; however, there is a significant difference between the maximum values. The maximum tidal level of the annual maxima method is 4.08 m, while that of the peaks over threshold method is 5.03 m. It demonstrates that the peaks over threshold method is more effective at obtaining extreme rainfall samples that will encounter a higher tide level.

### 4.2. Selection of Best-Fit Marginal Distributions

One of the most important and outstanding advantages of a copula function is that it can model the dependence structure independently of the choice of the marginal laws, and any marginal distribution can be constructed into a joint distribution using a copula function. Therefore, the marginal distributions of extreme precipitation and the corresponding high tide level can be estimated separately.

Five distributions were used to fit the marginal distributions of the extreme precipitation series and the corresponding high tide level series obtained using the two sampling methods (Figure 4). It can be seen that the Weibull distribution has a poor fitting effect at small observational values, while the other distributions have a satisfactory fitting effect on extreme precipitation and the corresponding high tide level.

To select the best-fit marginal distribution, four test methods (i.e., KS, CvM, *AIC*, and *BIC*) were applied to calculate the goodness-of-fit test statistics of extreme precipitation and the corresponding high tide level. The results of the goodness-of-fit test statistics are listed in Table 2. For the extreme rainfall series, all the goodness-of-fit statistics are in favor of the GEV distribution, it is the only distribution with three parameters, and thus it is preferred for the prediction of rainfall in different return periods. For the corresponding high tide level series, gamma distribution is selected as the optimal marginal distribution.

Although the marginal distributions of extreme rainfall and high tide level series obtained using the two sampling methods are identical, their parameters are different (Table 3). For the extreme rainfall series, the scale parameter of the annual maxima method is larger than that of the peaks over threshold method, whereas the converse is true for the corresponding high tide level series. It indicates that the marginal distribution of the extreme rainfall series sampled using the peaks over threshold method is more concentrated, whereas the marginal distribution of the corresponding high tide level series sampled using the annual maxima method is more concentrated.

On the basis of the optimal marginal distribution function, the design rainfall and tide level of the two sampling methods were estimated in different return periods, as shown in Figure 5. It can be seen that for the rainfall series, the design rainfall between the two sampling methods in different return periods is different, but the difference is small. When the return period is smaller than 132 years, the design value of the annual maxima method is larger than that of the peaks over threshold method, and with the increase in the return period, the difference in the design rainfall between the two sampling methods decreases. When the return period is greater than 132 years, the design value of the peaks over threshold method is greater than that of the annual maxima method, and the difference in the design rainfall between the two sampling methods widens with the increase in the return period. When the return period is 200 years, the difference is 19.83 mm. For the tidal level series, the design value of the peaks over threshold method is larger than that of the annual maxima method, and the difference is between 0.34 and 0.45 m. Therefore, in the planning and design of urban flood and tide control measures, for univariate design rainfall, when the return period is smaller than 132 (greater than 132) years, it is safer to use the annual maxima (peaks over threshold) method. For a univariate design tide level, the peaks over threshold method is generally safer.

### 4.3. Selection of Best-Fit Copula Function

The Kendall, Pearson, and Spearman correlation coefficients between the extreme rainfall and corresponding high tide level obtained using the two sampling methods are listed in Table 4. Although the values of the three correlation coefficients are reasonably small, they are all positive. The significance level (0.05) test indicated a significant and positive correlation between the extreme precipitation and corresponding high tide level, and that the correlation of the peaks over threshold method was greater than that of the annual maxima method. Therefore, copula functions could be used to construct the joint distribution of extreme precipitation and high tide level.

In this study, three Archimedean copulas (i.e., Clayton, Frank, and Gumbel copulas) were applied to select the best-fit copula. The empirical and theoretical cumulative distribution function (CDF) values based on the fitted copulas, presented in Figure 6 for comparison, indicate that all three copulas could be applied satisfactorily with the two sampling methods. In order to select the best-fit marginal distribution, the KS and CvM test, *AIC* and *BIC* criteria were calculated for these copulas with the smallest values then selected. The parameters of each copula were also calculated using the maximum likelihood method (Table 5). On the basis of these results, of the three copulas considered, the Clayton copula was selected as the best-fit copula for describing the dependence between the extreme rainfall and corresponding high tide level obtained using the two sampling methods.

According to the selection results of marginal distribution and the copula functions, the estimated joint distribution functions for extreme rainfall and the corresponding high tide level obtained using the two sampling methods can be presented as follows:(14)$$F_{AC}(r,t)={\left(F_{A}{\left(r\right)}^{-0.09}+F_{A}{\left(t\right)}^{-0.09}-1\right)}^{-1/0.09}$$
(15)$$F_{PC}(r,t)={\left(F_{P}{\left(r\right)}^{-0.208}+F_{P}{\left(t\right)}^{-0.208}-1\right)}^{-1/0.208}$$
where $F_{AC}(r,t)$ is the joint distribution function of the annual maxima method, $F_{A}(r)$ and $F_{A}(t)$ are the GEV and gamma marginal distributions for extreme rainfall and high tide level, respectively, obtained using the annual maxima method; $F_{PC}(r,t)$ is the joint distribution function of the peaks over threshold method, $F_{P}(r)$ and $F_{P}(t)$ are the GEV and gamma marginal distributions for extreme rainfall and high tide level, respectively, obtained using the peaks over threshold method.

Joint probability contours for extreme rainfall and corresponding high tide levels obtained using the two sampling methods are shown in Figure 7. The contour lines illustrate the joint probabilities of not exceeding both variables, and the joint probabilities for different return periods can be estimated using these iso-probability curves.

### 4.4. Joint Risk Probability Analysis

A risk analysis of natural disasters is generally expressed as an annual return period, which specifies the time interval between events of a similar magnitude. This definition does not mean that an event of the specified magnitude will only occur once in the interval of the return period, but it is a means of quantifying the probability that such an event will occur in any given year regardless of when the last similar event occurred. The return period of a single hydrological variable is uniquely determined, while there are several types of return period to analyze the joint risk in a bivariate case using the best-fit copula.

In this study, the bivariate joint return periods of “AND”, “OR”, and Kendall were used to analyze the joint risk of extreme rainfall and corresponding high tide level obtained using the two sampling methods. The joint exceeding probabilities of different combinations of return periods can be calculated based on the best-fit marginal distribution and copula functions, as shown in Table 6. It can be seen that for different combinations of return periods, the joint exceeding probabilities of “AND” cases were the smallest, those of “OR” cases were the largest, and those of Kendall cases were intermediate.

As an example, when the univariate return period of the annual maxima method is 10 years, the probability of a simultaneous occurrence of extreme rainfall exceeding 254.030 mm and the tide level exceeding 3.724 m is 1.08% per year, and the probability of only the extreme rainfall or the tide level exceeding the threshold occurrence is 18.92% per year in the Shenzhen River Basin. Thus, the “AND” and “OR” joint return periods, which can be calculated using Equations (8) and (9), are 92.575 and 5.285 years, respectively. The “AND” joint return period is much larger than the univariate return period, while the “OR” joint return period is nearly half of the univariate return period. It clearly indicates that if the positive dependency in the Shenzhen River Basin between extreme rainfall and the corresponding high tide level is ignored, the probability of simultaneous extreme events will be underestimated. The Kendall’s return period, which can be calculated using Equation (12), is 48.211 years, which is greater than the “OR” joint return periods and less than the “AND” joint return periods. This fully shows that the Kendall’s return period can avoid the overestimation or underestimation of the dangerous region according to the contour lines of the joint exceeding probabilities, and could more reasonably express the risk of combined extreme events.

For the different sampling methods, the joint exceeding probabilities of “AND” and Kendall cases for the peaks over threshold method are larger than those for the annual maxima method for the same univariate return period, whereas the converse is true for “OR” cases. The “AND”, “OR”, and Kendall joint return periods for different sampling methods are illustrated in Figure 8 for comparison. Generally, the joint return periods for the peaks over threshold method are larger than those for the annual maxima method. Then, the design rainfall and tide levels of different joint return periods for the different sampling methods can be calculated based on Figure 8.

In the planning and designing of flood and tide control engineering, the scale of the project is determined according to the return period of the extreme events. Therefore, the design rainfall and tide level corresponding to the return period are crucial for the safe operation of any project. Rainfall and tide levels with the same design frequency can be calculated according to the inverse function of the marginal distribution for a given joint return period.

Table 7 shows the design rainfall and tide levels of different types of joint return periods for the different sampling methods. It can be seen that for a specific sampling method (e.g., the annual maxima method), under the same joint return period (e.g., 100 years), the design values of rainfall and tide level corresponding to an “OR” joint return period (452.100 mm, 4.138 m) are greater than those corresponding to a Kendall joint return period (275.736 mm, 3.787 m), and that both are greater than those corresponding to an “AND” joint return period (256.263 mm, 3.731 m). The differences in the design values for the different type of joint return period are mainly attributable to the different ways of defining the hazardous area of extreme events. The “AND” joint return period narrows the area of a dangerous event when defining the dangerous area. Thus, the probability of the occurrence of a dangerous event becomes smaller under the combination of the same design values of rainfall and tide level, and the calculated joint return period is larger. In other words, under the same joint return period level, the design values of rainfall and tide level based on the “AND” joint return period are smaller. Similarly, the “OR” joint return period expands the area of a dangerous event when defining the dangerous area, which makes the calculated design rainfall and tide level values larger. The Kendall joint return period is more rigorous than the “AND” and “OR” joint return period when defining the dangerous area, avoids the overestimation and overestimation of the area of dangerous events, and can reasonably describe the probability of multivariate hydrological events in general.

For different sampling methods, under the same joint return period (e.g., 100 years), the design rainfalls corresponding to the “AND” and Kendall joint return period of the annual maxima method (256.263 mm, 275.736 mm) are larger than those of the peaks over threshold method (169.671 mm, 185.222 mm). However, the design tide levels corresponding to the “AND” and Kendall joint return period of the peaks over threshold method are larger than those of the annual maxima method. For the “OR” joint return period, the design tide level of the peaks over threshold method is slightly higher than that of the annual maxima method. The design rainfall of the peaks over threshold method is less than that of the annual maxima method when the joint return period is smaller than 100 years, but it is larger when the return period is greater than or equal to 100 years. The main reason for the differences in the design values for the same type of joint return period with different sampling methods is that the marginal distribution parameters of extreme rainfall and the corresponding high tide level are different for the different sampling methods.

As stated above, in the planning and designing of flood and tide control engineering, if a dangerous event is defined as one in which heavy rainfall and high tide level events occur simultaneously, the “AND” joint return period based on the annual maxima method should be adopted for the design rainfall, and the peaks over threshold method should be adopted for the design tide level. However, if a dangerous event is defined as one in which only a heavy rainfall or a high tide level event occurs, the “OR” joint return period should be adopted. For design rainfall, if a joint return period is smaller than 100 years, the annual maxima method is safer, whereas it is safer to use the peaks over threshold method for a return period of greater than or equal to 100 years. For the design tide level, the peaks over threshold method is safer. Generally, if there are no specific application requirements, we recommend the use of the Kendall joint return period based on the annual maxima method for risk analysis.

## 5. Conclusions

In this study, extreme value sampling methods, goodness-of-fit test methods and bivariate copula functions were used to quantitatively evaluate the joint risk of extreme rainfall and high tide level in the Shenzhen River Basin. The annual maxima and peaks over threshold methods were applied to construct extreme rainfall and the corresponding high tide level series on the basis of daily rainfall data from the Shenzhen Reservoir rainfall station and tide data from the Chiwan tide station acquired in the period 1965–2017. The extreme rainfall samples obtained by the peaks over threshold sampling method encountered more and higher tide levels, with an annual mean occurrence of 6.45 times.

According to the results of goodness-of-fit test, it was determined that GEV distribution and gamma distribution were the most suitable marginal distributions of extreme rainfall and the corresponding high tide level series obtained by using the two sampling methods, respectively, but that the parameters of the marginal distributions for the two sampling methods were different. Clayton copula was determined as the best-fit copula that was used for the joint risk analysis in this study.

A comparison of different types of joint return periods and univariate return periods of the same sampling method clearly indicated that if the positive dependency in the Shenzhen River Basin between extreme rainfall and corresponding high tide level were ignored, the probability of simultaneous extreme events would be underestimated. If a dangerous event is defined as one in which heavy rainfall and high tide level events occur at the same time, for design rainfall, the “AND” joint return period based on the annual maxima method should be adopted. If a dangerous event is defined as one in which only heavy rainfall or a high tide level event occurs, the “OR” joint return period should be adopted. For design rainfall, if a joint return period has a value smaller than 100 years, the annual maxima method is safer, whereas it is safer to use the peaks over threshold method for a return period with the value of greater than or equal to 100 years. For the design tide level, the peaks over threshold method is safer regardless of the sampling methods and the types of joint return periods. Generally, if there are no specific application requirements, the Kendall joint return period based on the annual maxima method is recommended for joint risk analysis.

This study could provide support for urban flood as well as tide control planning and design in the Shenzhen River Basin. With ongoing global climate change, the frequency and intensity of extreme rainfall events in different regions will change, the sea level will rise significantly, and the dependence between extreme rainfall and high tide level will also change. Therefore, a joint risk analysis of extreme precipitation and high tide level under the scenario of a changing environment is vitally important for the future safe development of cities, which is a subject that will be the focus of following studies.

## Figures and Tables

**Figure 1 ijerph-20-03605-f001:**
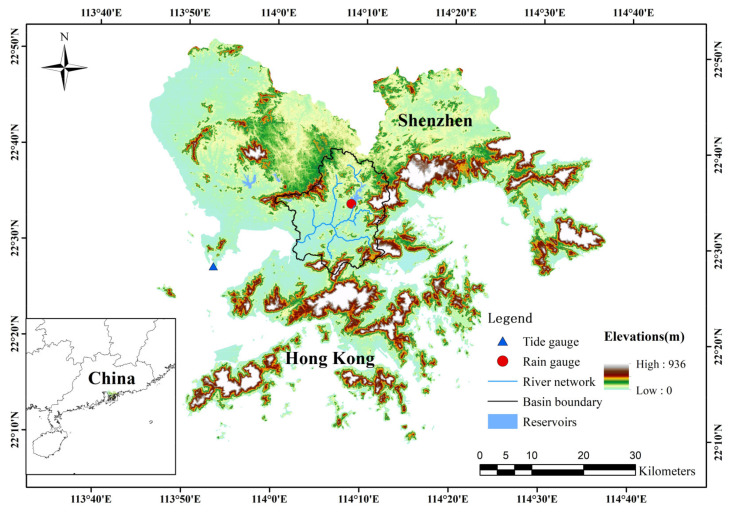
Locations of hydrologic gauges in the case study area.

**Figure 2 ijerph-20-03605-f002:**
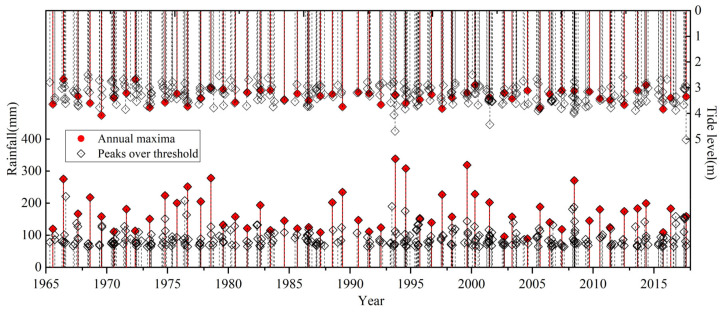
Extreme rainfall and corresponding high tide level series obtained using two sampling methods.

**Figure 3 ijerph-20-03605-f003:**
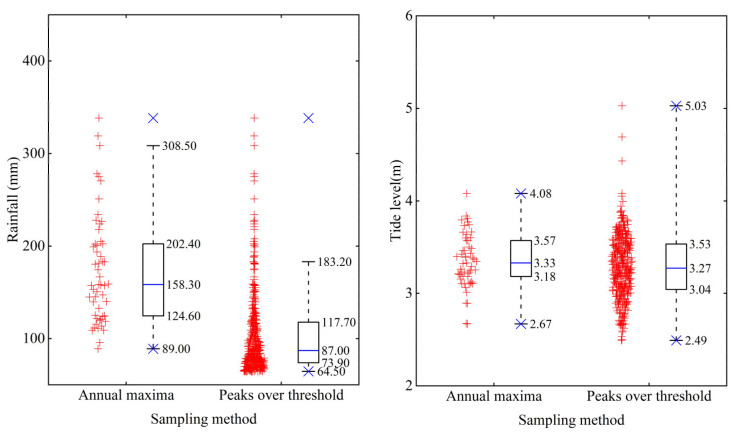
Fundamental statistical characteristics of the extreme rainfall and corresponding high tide series obtained using the two sampling methods. Scattered red crosses are the measured values, and the boxplot shows the statistical characteristic values. The lower and upper boundaries of the box represent the 25% and 75% quantiles, respectively. The blue solid line in the box is the median, the lines at both ends of the dotted line are the upper and lower limits of the normal values, and the blue “×” represents the maximum and minimum values of the sequence.

**Figure 4 ijerph-20-03605-f004:**
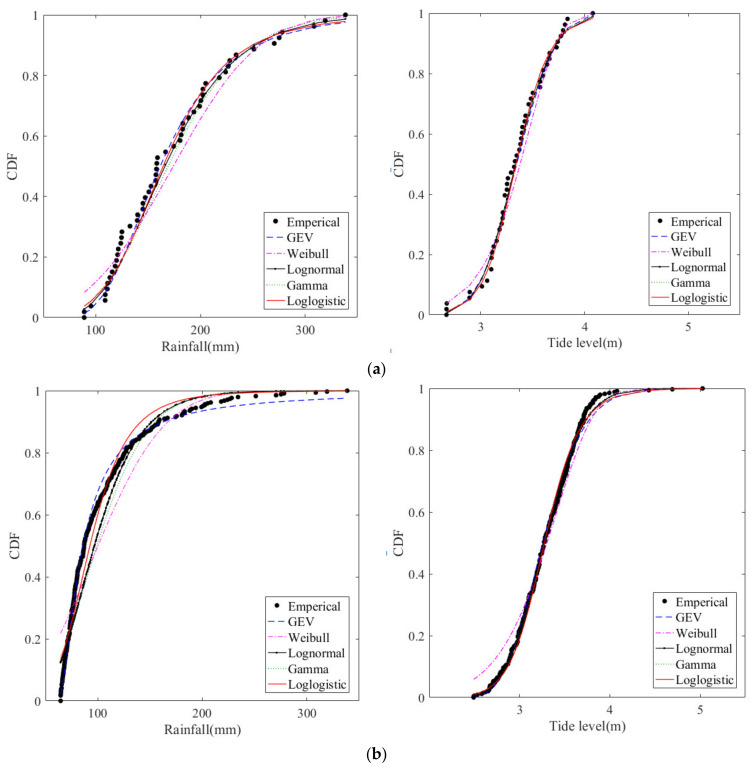
Comparison of cumulative distribution functions (CDF) of extreme rainfall and tide level series obtained using two sampling methods. (**a**) Annual maxima. (**b**) Peaks over threshold.

**Figure 5 ijerph-20-03605-f005:**
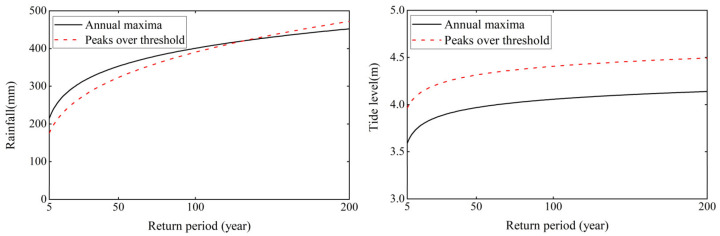
Comparison of the design rainfall and tide level of the two sampling methods.

**Figure 6 ijerph-20-03605-f006:**
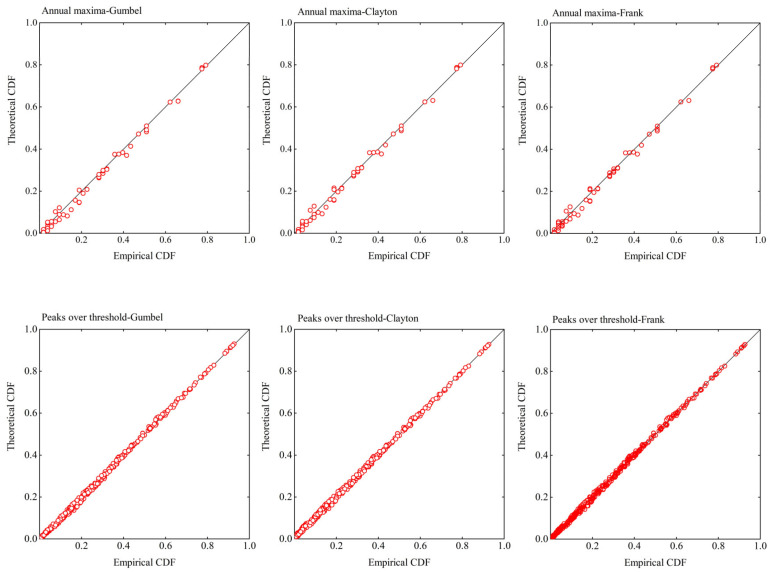
Comparison of empirical and theoretical cumulative distribution function values based on the fitted copulas.

**Figure 7 ijerph-20-03605-f007:**
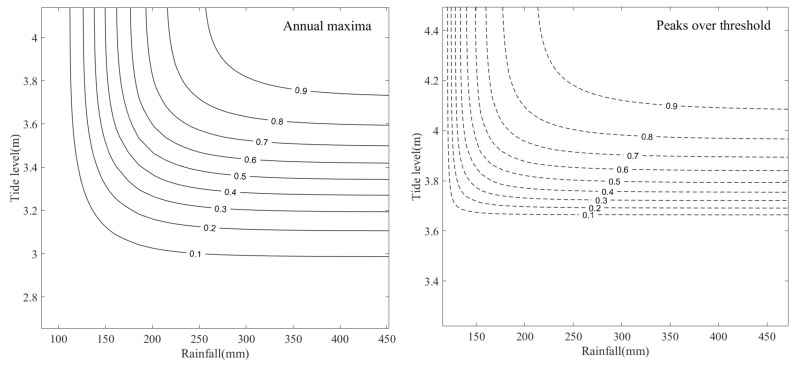
Contour lines of joint exceeding probabilities calculated using GEV and gamma marginal distributions for extreme rainfall and corresponding high tide levels obtained using two sampling methods.

**Figure 8 ijerph-20-03605-f008:**
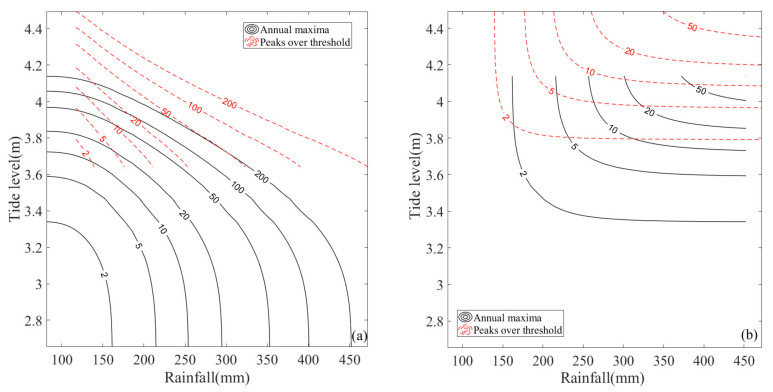
Comparison of the joint return periods for different sampling methods, (**a**) “AND” cases, (**b**) “OR” cases, and (**c**) Kendall cases.

**Table 1 ijerph-20-03605-t001:** Common bivariate Archimedean copula functions.

Copula	Generator ϕ	$C\left(\mu ,\nu \right)$	The Relationships between θ and Kendall τ	Parameter θ
Gumbel	$\phi_{\theta }(t)={\left(-\ln t\right)}^{\theta }$	$\exp\left\{-{\left[{\left(-\ln\mu \right)}^{\theta }+{\left(-\ln\nu \right)}^{\theta }\right]}^{1/\theta }\right\}$	$\tau =1-\frac{1}{\theta }$	θ∈[1,∞)
Clayton	$\phi_{\theta }(t)=t^{-\theta }-1$	${\left(\mu^{-\theta }+\nu^{-\theta }-1\right)}^{{}_{-1/\theta }}$	$\tau =\frac{\theta }{2+\theta }$	θ∈(0,∞)
Frank	$\phi_{\theta }(t)=-\ln(\frac{e^{-\theta t}-1}{e^{-\theta }-1})$	$-\frac{1}{\theta }\ln\left[1+\frac{\left(e^{-\theta \mu }-1\right)\left(e^{-\theta \nu }-1\right)}{e^{-\theta }-1}\right]$	$\tau =1+\frac{\theta }{4}\left[\frac{1}{\theta }\int_{0}^{\theta }\frac{t}{\exp(t)}dt-1\right]$	$\theta \in R\backslash \left\{0\right\}$

**Table 2 ijerph-20-03605-t002:** Goodness-of-fit test statistics for marginal distributions.

Sampling Method	Variable	Marginal Distribution Function	KS	CvM	*AIC*	*BIC*
Annual maxima	Rainfall	GEV	0.092	0.047	573.364	579.275
		Weibull	0.121	0.152	586.148	590.089
		Lognormal	0.092	0.050	576.258	580.199
		Gamma	0.102	0.076	578.402	582.343
		Loglogistic	0.092	0.056	579.108	583.049
	Corresponding tide level	GEV	0.065	0.042	24.191	29.742
		Weibull	0.094	0.131	27.125	30.826
		Lognormal	0.056	0.034	22.593	26.293
		Gamma	0.055	0.030	22.281	25.982
		Loglogistic	0.072	0.036	22.679	26.380
Peaks over threshold	Rainfall	GEV	0.051	0.148	3241.440	3252.087
		Weibull	0.200	2.528	3539.980	3547.078
		Lognormal	0.129	1.295	3385.420	3392.518
		Gamma	0.146	1.819	3438.700	3445.798
		Loglogistic	0.116	0.681	3375.400	3382.498
	Corresponding tide level	GEV	0.049	0.111	253.690	264.004
		Weibull	0.078	0.489	332.382	339.258
		Lognormal	0.035	0.057	244.704	251.580
		Gamma	0.033	0.043	244.152	251.028
		Loglogistic	0.042	0.103	248.890	255.766

**Table 3 ijerph-20-03605-t003:** CDFs of marginal distributions and their associated parameters.

Variable	Sampling Method	Marginal Distribution	CDF	Parameters
Rainfall	Annual maxima	GEV	$F(x)=\exp\left[-{\left(1+\xi \frac{x-\mu }{\sigma }\right)}^{{}^{{}^{-1/\xi }}}\right]$	location μ = 145.74 scale σ = 42.25 shape ξ = 0.11
	Peaks over threshold			location μ = 79.05 scale σ = 16.33 shape ξ = 0.29
Corresponding tide level	Annual maxima	Gamma	$F(x)=\int_{0}^{x}\frac{x^{\beta -1}}{\alpha^{\beta }\Gamma (\beta )}e^{-x/\alpha }dx$	scale α = 0.02 shape β = 135.08
	Peaks over threshold			scale α = 0.04 shape β = 90.28

**Table 4 ijerph-20-03605-t004:** Correlation coefficients between extreme rainfall and tide level series.

Correlation Coefficient	Pearson r	Kendall τ	Spearman ρ
Annual maxima	0.026	0.042	0.050
Peaks over threshold	0.084	0.072	0.107

**Table 5 ijerph-20-03605-t005:** Parameters and goodness-of-fit test statistics for Archimedean copulas.

Sampling Method	Copula	Parameter *θ*	Kendall’s tau	KS	CvM	*AIC*	*BIC*
Annual maxima	Gumbel	1.001	0.001	0.050	0.025	−407.004	−405.033
	Clayton	0.090	0.043	0.040	0.018	−425.342	−423.372
	Frank	0.271	0.030	0.044	0.020	−419.439	−417.469
Peaks over threshold	Gumbel	1.062	0.058	0.027	0.021	−3311.564	−3307.729
	Clayton	0.208	0.094	0.019	0.017	−3316.723	−3312.888
	Frank	0.713	0.079	0.023	0.021	−3312.622	−3308.787

**Table 6 ijerph-20-03605-t006:** Joint exceeding probabilities of different combinations of return periods.

Sampling Method	RP	Rainfall (mm)	Tide Level (m)	Probability of Exceeding Threshold		
				AND	OR	Kendall
Annual maxima	5	214.810	3.590	0.042818	0.357182	0.078699
	10	254.030	3.724	0.010802	0.189198	0.020742
	20	294.918	3.838	0.002713	0.097287	0.005318
	50	353.046	3.968	0.000435	0.039565	0.000864
	100	400.811	4.057	0.000109	0.019891	0.000217
	200	452.311	4.139	0.000027	0.009973	0.000054
Peaks over threshold	5	176.244	3.963	0.046367	0.353633	0.083989
	10	210.849	4.078	0.011832	0.188168	0.022549
	20	252.989	4.184	0.002989	0.097011	0.005837
	50	323.273	4.314	0.000481	0.039519	0.000953
	100	390.265	4.406	0.000121	0.019879	0.000240
	200	472.139	4.493	0.000030	0.009970	0.000060

**Table 7 ijerph-20-03605-t007:** Design rainfall and tide levels of different joint return periods for the different sampling methods.

Type	Joint RP (year)	Annual Maxima			Peaks over Threshold		
		Univariate RP (year)	Rainfall (mm)	Tide Level (m)	Univariate RP (year)	Rainfall (mm)	Tide Level (m)
AND	5	2.289	169.913	3.387	0.951	115.657	3.627
	10	3.256	190.596	3.489	1.352	126.447	3.709
	20	4.624	210.418	3.573	1.919	138.143	3.783
	50	7.337	236.383	3.667	3.043	155.264	3.874
	100	10.396	256.263	3.731	4.311	169.671	3.937
	200	14.72	276.560	3.789	6.103	185.513	3.997
OR	5	9.427	250.646	3.714	9.906	210.330	4.077
	10	19.442	293.203	3.833	19.906	252.674	4.183
	20	39.449	337.498	3.936	39.906	304.217	4.283
	50	99.452	400.418	4.056	99.906	390.165	4.406
	100	199.454	452.100	4.138	199.906	472.078	4.493
	200	399.454	507.818	4.216	399.906	572.186	4.578
Kendall	5	3.021	186.306	3.469	1.313	125.524	3.702
	10	4.403	207.668	3.562	1.881	137.447	3.779
	20	6.345	228.187	3.639	2.683	150.372	3.850
	50	10.189	255.105	3.727	4.273	169.296	3.936
	100	14.516	275.736	3.787	6.066	185.222	3.996
	200	20.634	296.815	3.842	8.600	202.735	4.054

## Data Availability

Not applicable.

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
