# Peer review of "Joint Risk Analysis of Extreme Rainfall and High Tide Level Based on Extreme Value Theory in Coastal Area"

_ijerph, 2023, doi:10.3390/ijerph20043605_

Round 1

Reviewer 1 Report

According to the author's research, it is crucial for the future safe growth of cities to conduct a joint risk analysis of extreme precipitation and high tide levels under the scenario of a changing environment. They used bivariate copula functions in their study to quantitatively assess the combined danger of extremely heavy rain and high tide in the Shenzhen River Basin (China). The study's findings appear to be valuable for society at first glance, and its methodology was genuine and accurate. According to the author's research, excessive rainfall and high tidal levels are the main causes of casualties in many Chinese cities. In this context, this study's methods and findings will be useful in preventing further catastrophic situations in China. There are some significant proposals that could be helpful for China and the rest of the world but are missing from the document.

Finally, include a couple more recent references that are relevant to your subject.

Reviewer 2 Report

Reviewer compliments authors on their thorough investigation as the analysis of joint risk analysis of extreme rainfall and high tide telvel is of great importance for coastal engineering. However, the authors should consider the following remarks in their next revision.

General remarks

1. It would be beneficial to have a paragraph (in the Introduction section) with research question(s) and hypotheses.
2. Some tables are split across pages and should be prepared for splitting if this split was intentional.

Specific remarks

1. In Table 2, in column 2 (named "variable"), it is not clear to which row particular variable refers to. Authors could add a horizontal line to split the part of the table for "Rainfall" from the "Corresponding tidal level" rows.
2. In lines 446-449, authors state that "AND" and "OR" return periods are 92.575 and 5.285 years nut they cannot be found in any additional table. Authors should consider computing and storing such results in a table in the main text or the appendix and reference them.

Reviewer 3 Report

See the comments attached.
